# Characterization and Function Analysis of miRNA Editing during Fat Deposition in Chinese Indigenous Ningxiang Pigs

**DOI:** 10.3390/vetsci11040183

**Published:** 2024-04-22

**Authors:** Jiayu Lv, Fang Yang, Yiyang Li, Ning Gao, Qinghua Zeng, Haiming Ma, Jun He, Yuebo Zhang

**Affiliations:** 1College of Animal Science and Technology, Hunan Agricultural University, Changsha 410128, China; lvjiayu724@163.com (J.L.); yf621829@stu.hunau.edu.cn (F.Y.); 18236635760@163.com (Y.L.); gaon@hunau.edu.cn (N.G.); chuweixiang168168@163.com (Q.Z.); mahaiming@hunau.edu.cn (H.M.); 2Key Laboratory of Livestock and Poultry Resources (Pig) Evaluation and Utilization, Ministry of Agriculture and Rural Affairs, Changsha 410000, China

**Keywords:** Ningxiang pigs, miRNA editing, high-throughput sequencing, fat deposition

## Abstract

**Simple Summary:**

In order to gain insight into the molecular mechanism of porcine fat deposition, this study reported, for the first time, miRNA editing in the adipose tissue of Ningxiang pigs. We performed bioinformatics analyses, such as developmental stage-specific site screening, target gene prediction, and functional enrichment analysis, to obtain the functional miRNA editing sites associated with fat deposition; we found that miR-497 editing might inhibit fat deposition in pigs through retargeting genes. These findings not only enhance our understanding of the functional roles and mechanisms of miRNA editing in adipose development but also hold significant importance for improving the lean meat percentage of Ningxiang pigs and promoting their industrial development.

**Abstract:**

This study aimed to identify active miRNA editing sites during adipose development in Ningxiang pigs and analyze their characteristics and functions. Based on small RNA-seq data from the subcutaneous adipose tissues of Ningxiang pigs at four stages—30 days (piglet), 90 days (nursery), 150 days (early fattening), and 210 days (late fattening)—we constructed a developmental map of miRNA editing in the adipose tissues of Ningxiang pigs. A total of 505 miRNA editing sites were identified using the revised pipeline, with C-to-U editing types being the most prevalent, followed by U-to-C, A-to-G, and G-to-U. Importantly, these four types of miRNA editing exhibited base preferences. The number of editing sites showed obvious differences among age groups, with the highest occurrence of miRNA editing events observed at 90 days of age and the lowest at 150 days of age. A total of nine miRNA editing sites were identified in the miRNA seed region, with significant differences in editing levels (*p* < 0.05) located in ssc-miR-23a, ssc-miR-27a, ssc-miR-30b-5p, ssc-miR-15a, ssc-miR-497, ssc-miR-15b, and ssc-miR-425-5p, respectively. Target gene prediction and KEGG enrichment analyses indicated that the editing of miR-497 might potentially regulate fat deposition by inhibiting adipose synthesis via influencing target binding. These results provide new insights into the regulatory mechanism of pig fat deposition.

## 1. Introduction

Ningxiang pigs are one of the excellent local pig breeds in China and have the characteristics of early maturity and easy fattening, wide adaptability, and tasty meat, and thus, are popular among consumers [1]. However, it has characteristics such as high fat percentage and low lean meat percentage [2], low feed utilization efficiency, slow growth, and low meat yield, which seriously affect the efficiency and benefits of Ningxiang pig farming, representing a key bottleneck for the development of the Ningxiang pig industry.

An in-depth analysis of the molecular mechanism of fat deposition is a prerequisite for regulating the rational deposition of fat in a targeted manner. In 1986, Benne et al. [3] reported, for the first time, RNA editing, an important and specific way of post-transcriptional modification in eukaryotic genes, which alters the genetic information conveyed by RNAs. RNA editing that occurs in miRNAs is usually referred to as miRNA editing. miRNAs [4] are endogenous non-coding RNAs consisting of about 22 nucleotides. They post-transcriptionally regulate messenger RNAs (mRNA) by binding to the 3′ UTR of the mRNA by partial sequence complementarity, mainly in the seed sequence (nucleotide positions 2–8 from the 5′ end of the miRNA) [5,6]. miRNAs are widely present in pig adipose tissue and participate in regulating multiple biological processes related to pig fat deposition. It has been shown that miRNAs regulate adipocyte differentiation and function by targeting specific mRNAs, thereby affecting fat deposition in pigs [7]. During adipocyte differentiation, ssc-miR-7134-3p inhibits the expression of *MARK4* protein, thereby affecting adipocyte differentiation and fat deposition [8]. miR-34a affects fat deposition by targeting *LEF1* [9], and miR-503 inhibits adipocyte differentiation by targeting *MafK*, leading to a reduction in back fat thickness [10].

Research on miRNA editing has never stopped since it was first reported in 2004 [11]. A large number of miRNA editing studies have focused on cancer, such as the downregulation of miR-17 [12] in melanoma stem cells, which inhibits the activity of melanoma stem cells and promotes cell differentiation. There is limited research on miRNA editing in fat deposition. Meadows et al. found that highly edited miRNAs are closely associated with adipogenic differentiation by analyzing different stages of mouse adipose tissue development [13]. However, whether miRNA editing affects fat deposition in pigs remains unclear.

Given the influence of miRNA editing on the targeting activity of host miRNAs and the significant role of miRNAs in pig fat deposition, this study utilized small RNA-seq technology to identify and analyze miRNA editing sites in the subcutaneous fat tissue of Ningxiang pigs at four different stages (N30D, N90D, N150D, and N210D). We aimed to construct a miRNA editing developmental atlas for the adipose tissue of Ningxiang pigs and screen the functional miRNA editing sites that impact pig fat deposition, which provide novel insights into unraveling the regulatory mechanisms of pig fat deposition.

## 2. Materials and Methods

### 2.1. Experimental Animals

The small RNA sequencing data used in this study were obtained from our previous study [14]. Briefly, subcutaneous adipose tissue samples from 12 Ningxiang boars at four different developmental stages—N30D (piglets), N90D (nursery), N150D (early fattening), and N210D (late fattening)—were analyzed using small RNA-seq. Each age group included three replicates labeled F1, F2, and F3 (all pigs were half-sibs). The pigs were provided by Dalong Animal Husbandry Technology Co. Ltd. in Hunan Province, China. These data are available at NCBI BioProject: PRJNA721288 (https://www.ncbi.nlm.nih.gov/bioproject/PRJNA721288/ (accessed on 5 March 2024)).

### 2.2. Filtration and Comparison of Sequencing Data

Sequencing data were filtered based on the following conditions: (1) removal of low-quality reads (where bases with a quality score Q ≥ 30 accounted for less than 95% of the entire read); (2) trimming of adapter sequences; (3) exclusion of reads that were longer than 28 bp or shorter than 15 bp. Subsequently, we aligned the filtered reads to the pig reference genome (Sus scrofa 11.1) using Bowtie [15] to obtain uniquely mapped reads with no more than one mismatch (parameters: -n 1 -e 50 -a -m 1 --best --strata --trim3 2). Given the common adenylation and uridylation modifications at the 3′ end of mature miRNAs [16,17], the 2 nt at the 3′ end were trimmed during alignment.

### 2.3. Identification and Characterization of miRNA Editing Sites

The method for identifying miRNA editing sites was based on the approach reported by Alon et al. [18], involving the following: (1) aligning filtered data to the pig reference genome and then to the known precursor miRNA (Pre-miRNA) sequences in the miRBase database (version 21) [19] to obtain the quantity of each of the four nucleotides at every position within all Pre-miRNA sequences; (2) performing mismatch detection based on the binomial test with Bonferroni-corrected *p* values ≤ 0.05 and a mismatch base quality phred score of ≥30 to identify miRNA editing candidates; (3) excluding known single nucleotide polymorphisms (SNPs) from the dbSNP database (release 150). Further selection of candidate miRNA editing sites included the following: (1) selecting sites present in at least three individuals; (2) excluding sites with multiple editing types; (3) considering only miRNA editing sites with a coverage of ≥10 reads.

The most abundant four types of miRNA editing sites (A-to-G, C-to-U, G-to-U, and U-to-C) and their flanking sequences (5 bp upstream and downstream) were analyzed for nucleotide preferences using the Two Sample Logos online tool (http://www.twosamplelogo.org/ (accessed on 5 March 2024)). RNA editing level was defined as the ratio of editing-type reads to the total reads detected at that site. Time series analysis of average editing levels across stages was conducted using the Mfuzz package [20]. Due to the comprehensive nature of current research on human miRNA editing, with A-to-G editing types accounting for the majority of identified editing sites in humans, we downloaded human A-to-G miRNA editing sites from the MiREDiBase database (https://ncrnaome.osumc.edu/miredibase (accessed on 5 March 2024)) [21]. Subsequently, we used the bedtools [22] software to extract 25 bp flanking sequences upstream and downstream of the editing sites. Following this, we utilized the makeblastdb command to construct a BLAST database for human miRNA editing sites. The flanking sequences of A-to-G miRNA editing sites extracted by bedtools were used as query sequences. We then employed BLASTN to search the human RNA editing site BLAST database, and miRNA editing sites for query sequences with an e-value of <0.001 and >85% consistency were regarded as conserved sites between humans and pigs.

### 2.4. Screening of Differential miRNA Editing Sites

REDITs [23] is a tool that uses a β-binomial model to identify differential editing sites. The tool takes into account both the variance in editing resulting from biological variability and the intrinsic inaccuracy arising from calculating editing from counting data such as RNA-seq. As a result, it exhibits greater power and lower false positives at and below the 5% false positive threshold compared to commonly used alternatives, such as the t-test, Wilcoxon’s rank-sum test, or pooled Fisher’s exact test. In this study, 6 pairwise comparisons (N30D vs. N90D, N30D vs. N150D, N30D vs. N210D, N90D vs. N150D, N90D vs. N210D, and N150D vs. N210D) were conducted to investigate the differences in editing levels between developmental stages.

### 2.5. Prediction of Target Genes and KEGG Functional Enrichment Analysis

In order to evaluate the impact of differentially edited sites within miRNA seed sequences on miRNA function, target genes were predicted for wild-type (WT) and edited-type (ET) miRNAs using the miRanda software [24] with the following parameters: -sc 140 -en -10 -scale 4. Subsequently, the Kyoto Encyclopedia of Genes and Genomes (KEGG) pathway enrichment analysis of these target genes was conducted using the OmicShare online platform (https://www.omicshare.com/tools (accessed on 5 March 2024)).

## 3. Results

### 3.1. Identification of miRNA Editing Sites

A total of 505 miRNA editing sites were identified, encompassing 10 nucleotide editing types: A-to-G, G-to-A, C-to-U, U-to-C, G-to-U, C-to-G, G-to-C, A-to-C, U-to-G, and A-to-U. C-to-U editing sites accounted for the majority, with 146 editing sites (Figure 1A). The distribution of non-redundant miRNA editing sites in the subcutaneous adipose tissue of Ningxiang pigs at different stages is shown in Figure 1B. Among them, the 90-day-old had the most miRNA editing events, while the 150-day-old had the least. There was a significant difference between N30D and N90D (*p* < 0.05), between N90D and N150D (*p* < 0.01), between N150D and N210D (*p* < 0.05), and between N30D and N150. The number of miRNA editing sites was also varied among different individuals at the same stage (Figure 1C). These results suggest that the distribution of miRNA editing sites is not only specific to different developmental stages but also varies among individuals within the same stage, indicating a spatiotemporal pattern of miRNA editing.

### 3.2. miRNA Editing Characteristic Analysis

In order to explore whether there was a base preference in the flank sequence of the miRNA editing sites, we analyzed the sequence patterns of the four most abundant miRNA editing sites (A-to-G, C-to-U, G-to-U, and U-to-C) obtained from Figure 1A. As shown in Figure 2A, there was no preference at 1 bp upstream of the A-to-G editing site, where G and C were preferred, but A and T were excluded at 1 bp downstream. As shown in Figure 2B, 1 bp upstream of the C-to-U editing site favored C, and T was rejected at 1 bp downstream. As shown in Figure 2C, there is no preference within 1 bp upstream and downstream of the G-to-U editing site, with the upstream 2 bp rejecting A, and the downstream 2 bp favoring A and rejecting C. As shown in Figure 2D, 1 bp upstream of the U-to-C editing site preferred G and rejected T, while there was no preference within 4 bp downstream, and G was preferred at 5 bp downstream.

In order to investigate the dynamic changes in the average editing levels across different stages during fat development (Figure 3), Mfuzz was used to categorize the average editing levels into four patterns (Cluster 1, Cluster 2, Cluster 3, and Cluster 4). For Cluster 1, the average editing levels exhibited a gradual increase from N30D to N90D and a rapid decrease followed by an increase from N90D to N210D. In Cluster 2, the average editing levels showed a rapid increase, decrease, and then increase again. Cluster 3 displayed a rapid decrease and increase in the average editing levels from N30D to N90D and from N90D to N210D. Meanwhile, for Cluster 4, the average editing levels rapidly increased from N30D to N90D, followed by a progressively slower decrease from N90D to N150D and N150D to N210D. These findings indicated that the miRNA editing levels during fat development might not change smoothly but rather undergo distinct transitions. This suggests a close association between fat development and miRNA editing.

In order to investigate the conservation of the identified miRNA editing sites, a cross-species analysis was performed between the A-to-G miRNA editing sites identified in Ningxiang pigs and those reported in humans. A total of 67 conserved sites were identified between the two species (Appendix A).

### 3.3. Identification of Differential Editing Sites

In order to select the miRNA editing sites with significantly different editing levels in the seed region, pairwise comparisons of the editing levels at each site were conducted across the 4 stages. The results showed that 37 sites differed in editing levels in only one comparison group, 17 sites differed in editing levels in three comparison groups, and eight sites differed in editing levels in all comparison groups; there were no shared differential editing level sites in 2, 4, and 5 comparison groups. (Figure 4A). Among the different comparison groups, there are 62 miRNA editing sites with significant differences in editing levels. After excluding the sites (located on sex chromosomes), 20 of these were located in the seed region (Table 1). In the analysis of the editing level of the 20 editing sites, it was found that there are 14 sites with differences in only one comparison group, four sites with differences in three comparison groups, and two sites with differences across all comparison groups (Figure 4B).

### 3.4. Target Gene Prediction and Functional Enrichment Analysis for A-to-G Editing Site Host

Among the 20 differential editing sites located in the seed region, there are nine editing sites that have the A-to-G type: 2_65308161_A-to-G, 2_65308350_A-to-G, 4_6952809_A-to-G, 4_6952808_A-to-G, 11_17757478_A-to-G, 12_52422400_A-to-G, 12_52422397_A-to-G, 13_100083195_A-to-G, and 13_31655056_A-to-G, which are respectively located in ssc-miR-23a, ssc-miR-27a, ssc-miR-30b-5p, ssc-miR-15a, ssc-miR-497, ssc-miR-15b, and ssc-miR-425-5p. Notably, ssc-miR-497_2 and ssc-miR-497_5 indicate that editing occurs in bases 2 and 5 of ssc-miR-497, and ssc-miR-30b-5p_5 and ssc-miR-30b-5p_6 indicate that editing occurs in bases 5 and 6 of ssc-miR-30b-5p. These miRNA editing sites led to significant changes in the target gene profiles of their host miRNAs. Specifically, the editing at these sites resulted in a loss of 2310 to 3591 target genes and, concurrently, a gain of 187 to 2078 new target genes (Figure 5). The comparative KEGG enrichment analysis of all wild-type (WT) and edited-type (ET) miRNA target genes revealed changes in the enriched pathways related to lipid metabolism, including a loss in the PI3K-Akt signaling pathway and two new pathways: the AMPK signaling pathway and insulin signaling pathway (Figure 6). The list of target genes for the 9 A-to-G miRNA editing sites by WT and ET miRNAs is shown in Appendix A. The KEGG enrichment analysis results of the WT and ET miRNA target genes of the 9 A-to-G miRNA editing sites are shown in Appendix A.

## 4. Discussion

Previous studies on miRNA editing have primarily focused on cancer. This study is the first to link pig fat deposition with miRNA editing, analyzing its impact on fat development. In this study, adipose tissue from pigs at 30 days (piglet stage), 90 days (nursery stage), 150 days (early fattening stage), and 210 days (late fattening stage) were selected for in-depth analysis. These periods are critical stages of growth and development in pigs, and their physiological and metabolic activities undergo significant transformations. Given that it has been demonstrated that miRNAs play a crucial role in the regulation of porcine adipose development, we hypothesized that miRNA editing may have an important impact on the regulation of fat deposition in Ningxiang pigs. There are fewer miRNA editing sites in the adipose tissue of Ningxiang pigs compared to humans [18]. Unexpectedly, C-to-U is the most common type in pigs, while A-to-G is the most common type in humans. However, the result of this study is highly consistent with the study of Wang et al. [25], which analyzed miRNA editing sites during pig sperm development.

The number of editing sites of miRNAs showed specificity across different developmental stages and varied among individuals at the same stage, indicating that the editing distribution of miRNAs is spatiotemporal-specific. This is similar to the developmental stage specificity of RNA editing events [26]. During adipose development, from 30 to 90 days of age, coinciding with rapid adipose tissue growth, miRNA editing might play a crucial role in target gene regulation, and the number of editing sites increases accordingly. As pigs transition to the early fattening stage (150 days), subcutaneous adipose development may stabilize, resulting in a decreased need for miRNA editing. The increase in miRNA editing events in the late fattening stage (210 days) may be related to metabolic demands as pigs enter this stage. These findings suggest that changes in miRNA editing may be linked to the needs of organismal tissues, aligning with previous comments about the importance of RNA editing increasing with organismal complexity [27]. miRNA editing might affect the life activities of animals by regulating target genes via changing the targeting and function of miRNAs [28,29].

Additionally, it may be too arbitrary to speculate on the relationship between adipose development and miRNA editing just by analyzing the distribution of miRNA editing sites, so we further analyzed the changes in editing levels at each stage. The dynamic changes in miRNA editing levels during adipose development showed a significant cyclic pattern, a finding that suggests a possible close association between fat deposition and miRNA editing in Ningxiang pigs. This cyclic change contrasts with the continuous rise in editing levels observed in brain development [30]. We hypothesize that this discrepancy may arise from tissue-specific regulatory requirements; while RNA editing may be involved in the differentiation and maturation of neural precursor cells during the early stages of brain development, miRNA editing during adipose development may be regulated by the demands of complex life activities that do not fully rely on sustained editing activity for fine regulation.

Although our research provides valuable insights based on sequencing data, direct tissue analysis has not yet been conducted to validate these findings. Future research will focus on validating these changes to gain a deeper understanding of the molecular mechanisms underlying fat deposition in Ningxiang pigs.

Furthermore, miRNA editing in the seed region can lead to changes in seed sequences, which, in turn, affect the targeting effect of host miRNAs. Therefore, this study screened miRNA editing sites in the seed region with significantly different editing levels. It has been found that there are numerous RNA editing events in pig fat tissue, with the majority of editing events being A-to-G edits, which might play an important role in pig fat deposition [31]. Furthermore, the miRNA editing sites that have been functionally characterized are almost exclusively of the A-to-G editing type [32,33]. Therefore, this study focuses on A-to-G editing sites to analyze the regulatory role of miRNA editing more accurately and specifically on fat deposition in Ningxiang pigs.

In the miRNA seed region, nine differentially edited A-to-G miRNA sites (2_65308161_A-to-G, 2_65308350_A-to-G, 4_6952809_A-to-G, 4_6952808_A-to-G, 11_17757478_A-to-G, 12_52422400_A-to-G, 12_52422397_A-to-G, 13_100083195_A-to-G, and 13_31655056_A-to-G) were ultimately selected, which were located in ssc-miR-23a, ssc-miR-27a, ssc-miR-30b-5p, ssc-miR-15a, ssc-miR-497, ssc-miR-15b, and ssc-miR-425-5p, respectively. These host miRNAs are all related to fat deposition. For instance, miR-23a promotes lipid accumulation [34]; several studies have reported that miR-27a affects apoptosis and insulin resistance in adipocytes [35,36,37]; miR-30b-5p regulates intracellular lipid metabolism by targeting PPARGC1 [38]; miR-15a regulates the differentiation of preadipocytes in Yanbian cattle by inhibiting the expression of ABAT [39]; miR-497 is involved in regulating fatty acid synthesis and affecting insulin resistance [40]; miR-15b participates in lipid synthesis [41]; and miR-425-5p inhibits the differentiation and proliferation of preadipocytes [42].

Additionally, a comparative KEGG enrichment analysis of all wild-type (WT) and edited-type (ET) miRNA target genes revealed the changes in the enriched pathways related to lipid metabolism, where the PI3K-Akt signaling pathway was lost, whereas the AMPK signaling pathway and insulin signaling pathway were newly enriched. The AMPK signaling pathway is typically considered a metabolic regulatory hub under conditions of energy expenditure, with its activation inhibiting fatty acid synthesis and promoting fatty acid oxidation, thereby reducing fat deposition [43,44,45,46]. Conversely, the PI3K-Akt signaling pathway promotes the growth, differentiation, and fatty acid synthesis of adipocytes, leading to increased fat deposition [47,48,49]. Therefore, miRNA editing may play a significant role in regulating fat deposition.

Of note, the edited ssc-miR-497_2 gained the most new target genes, and they were significantly enriched in a new pathway, the Wnt signaling pathway, which is associated with the regulation of fat deposition [50] (Appendix A). These findings suggest that miR-497 editing may introduce a novel regulatory role and function in fat deposition. Furthermore, studies have shown that the A-to-G editing site located within miR-497 (12_52422400_A-to-G) is conserved among humans, mice, kangaroos, rhesus monkeys, and pigs [51], which is of the highest editing level in our study. Therefore, this site was chosen as a key candidate.

A comparative analysis of the target genes significantly enriched in the lipid metabolism pathways revealed that the edited ssc-miR-497 loses target genes such as *SCD* [52], *PLAAT3* [53], *PNPLA6* [54], *ACSL6* [55], *ASAH2* [56], *CHPT1* [57], *FADS2* [58] and *ACSL4* [59], all of which promote fat synthesis and differentiation. Conversely, the newly acquired target genes are primarily involved in fat metabolism, including *PLA2G12A* [60], *LPGAT1* [56], *GGT1* [61], *HADH* [62], *MGLL* [63], *CPT1B* [64], etc. Therefore, we speculated that ssc-miR-497 editing may play an inhibitary role in regulating fat deposition in Ningxiang pigs. Based on this speculation, corresponding validation experiments will be conducted in animals or cells in the future.

Through the careful investigation of the physiological changes and fat development activities of pigs at different growth stages, the pig industry can obtain valuable information, which is very important for optimizing feeding management, improving feed formulas, strengthening disease control measures, and improving all aspects of pig production. Especially at the molecular level of fat development, it is of great significance to understand the characteristics of fat in pigs, such as Ningxiang pigs, for variety improvement and meat quality improvement. In this process, miRNA editing, as a new molecular regulation mechanism, shows its key role in regulating fat deposition and influencing many biological processes, and it may play a decisive role in improving pig production performance and genetic improvement. When compared with human beings, although the research on pig miRNA editing is still insufficient, further exploration in this field will undoubtedly bring new perspectives and profound understanding to pig genetics research, which deserves the common attention of academia and industry and more comparative research.

## 5. Conclusions

This study is the first to report on miRNA editing in pig adipose tissue, identifying and analyzing miRNA editing sites during fat development. We further selected functionally relevant miRNA editing sites and their target genes. According to our target gene annotation and functional enrichment analysis results, ssc-miR-497 editing may play an important role in regulating fat deposition in Ningxiang pigs. These results provide new insights into the regulatory mechanism of pig fat deposition.

## Figures and Tables

**Figure 1 vetsci-11-00183-f001:**
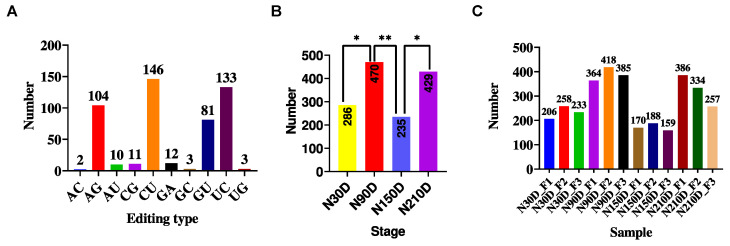
Distribution of miRNA editing sites. (**A**) Distribution of miRNA editing sites among editing types; (**B**) Distribution of miRNA editing sites among 12 samples; (**C**) Distribution of miRNA editing sites among stages. * indicates a significant difference between stages (*p* < 0.05); ** indicates an extremely significant difference between stages (*p* < 0.01).

**Figure 2 vetsci-11-00183-f002:**
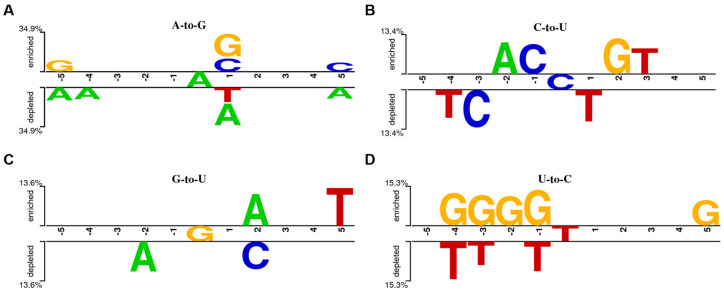
Characteristics of the upstream and downstream bases of the A-to-G (**A**), C-to-U (**B**), G-to-U (**C**), and U-to-C (**D**) miRNA editing sites. In the figure, the region from −5 to −1 is defined as the upstream region, while the region from 1 to 5 is defined as the downstream region.

**Figure 3 vetsci-11-00183-f003:**
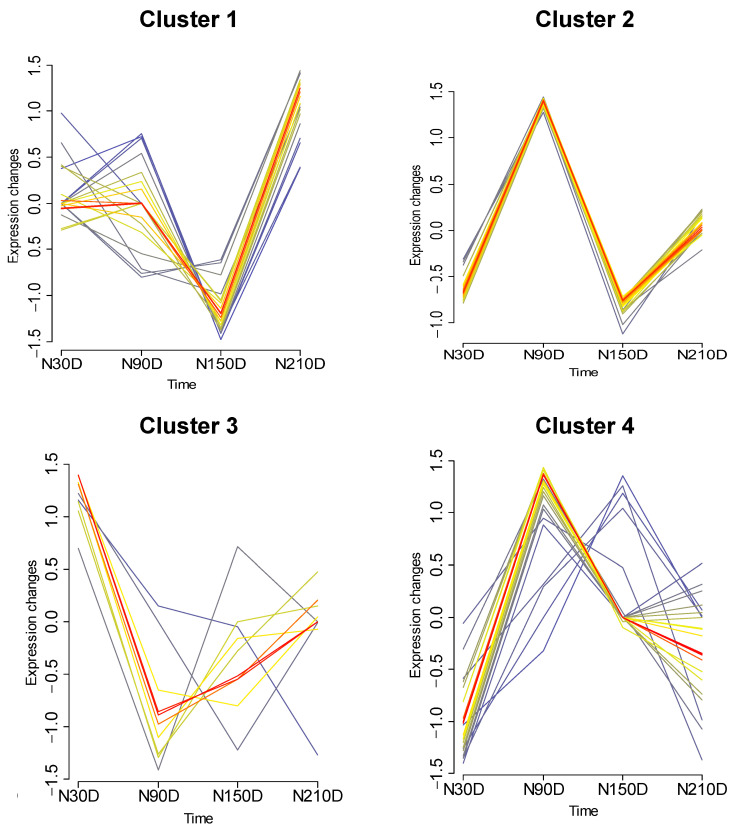
Time series analysis of average editing level among different stages.

**Figure 4 vetsci-11-00183-f004:**
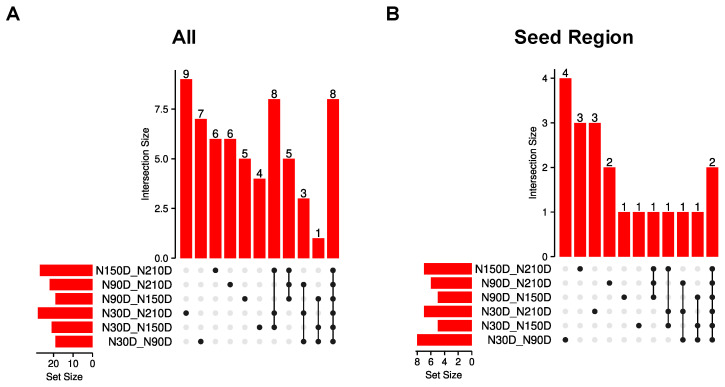
Distribution of miRNA editing sites with significant differences in editing levels; (**A**) distribution of differential editing sites among the compared groups; (**B**) distribution of differential editing sites in the seed region among the compared groups.

**Figure 5 vetsci-11-00183-f005:**
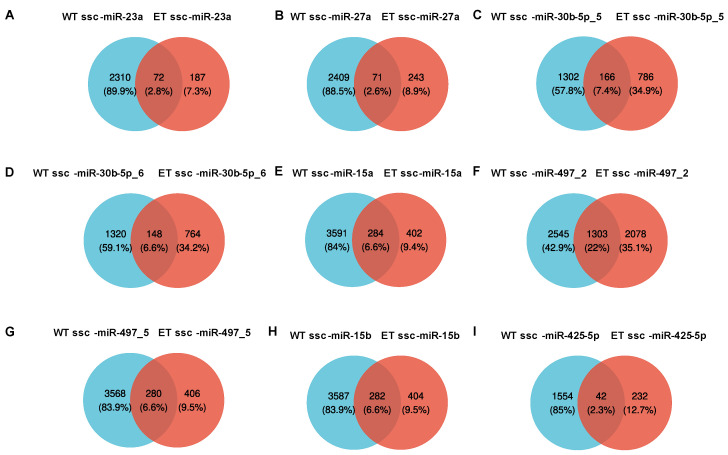
Target genes of wild-type (WT) and edited-type (ET) miRNAs. ssc-miR-23a (**A**), ssc-miR-27a (**B**), ssc-miR-30b-5p_5 (**C**), ssc-miR-30b-5p_6 (**D**), ssc-miR-15a (**E**), ssc-miR-497_2, (**F**) ssc-miR-497_5 (**G**), ssc-miR-15b (**H**), and ssc-miR-425-5p (**I**).

**Figure 6 vetsci-11-00183-f006:**
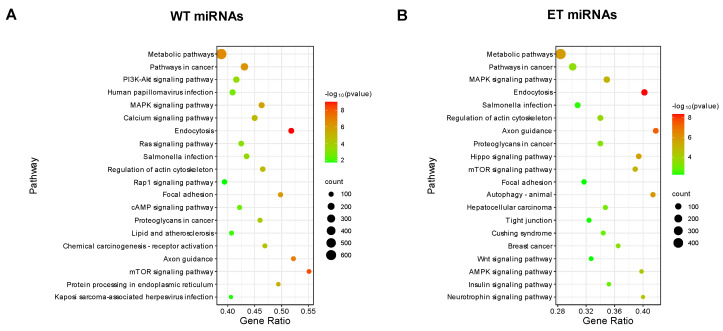
KEGG enrichment analysis results for miRNAs with A-to-G differential editing sites in the seed region. (**A**) KEGG enrichment analysis results for wild-type (WT) miRNAs; (**B**) KEGG enrichment analysis results for edited-type (ET) miRNAs.

**Table 1 vetsci-11-00183-t001:** Differential editing sites in the miRNA seed region.

Chromosome	Position	Mature miRNA Name	Position in miRNA	Editing Type
2	65308161	ssc-miR-23a	4	A-to-G
2	65308350	ssc-miR-27a	4	A-to-G
4	6952809	ssc-miR-30b-5p	5	A-to-G
4	6952808	ssc-miR-30b-5p	5	A-to-G
11	17757478	ssc-miR-15a	5	A-to-G
12	52422400	ssc-miR-497	2	A-to-G
12	52422397	ssc-miR-497	5	A-to-G
13	100083195	ssc-miR-15b	5	A-to-G
13	31655056	ssc-miR-425-5p	7	A-to-G
2	65308351	ssc-miR-27a	5	C-to-U
12	46211541	ssc-miR-423-5p	8	C-to-U
14	6520933	ssc-miR-320	6	C-to-U
1	224065570	ssc-miR-204	2	U-to-C
2	150580147	ssc-miR-145-5p	7	U-to-C
2	65308354	ssc-miR-27a	8	U-to-C
6	58332107	ssc-let-7e	6	U-to-C
12	45088852	ssc-miR-451	8	U-to-C
12	43337029	ssc-miR-193a-5p	5	U-to-C
13	189138833	ssc-miR-155-5p	2	U-to-C
15	120453426	ssc-miR-26b-5p	7	U-to-C

## Data Availability

No new data were created in this study. Data sharing is not applicable to this article.

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
