# Peer review of "Characterization and Function Analysis of miRNA Editing during Fat Deposition in Chinese Indigenous Ningxiang Pigs"

_vetsci, 2024, doi:10.3390/vetsci11040183_

Round 1

Reviewer 1 Report

Comments and Suggestions for Authors

Overall, the study provides valuable insights into miRNA editing sites during adipose tissue development in Ningxiang pigs, contributing to our understanding of fat deposition regulation in this breed. However, there are some areas that require clarification and improvement before publication in this journal.

    The presentation of research findings, especially the explanation of Figure 1, needs enhancement. Currently, the results lack sufficient elaboration, making it difficult for readers to comprehend, particularly those new to the topic. Additional explanations or annotations within Figure 2 could help clarify the findings.

    Figure 2 lacks clear markings, and it would be beneficial to include labeling of upstream and downstream regions by base pair (bp) for better comprehension. Clearer markings would assist readers in understanding the spatial context of the analyzed sequences.

    In Figure 4A, there appears to be a discrepancy regarding the four comparison groups mentioned in line 194 of the results section. This discrepancy requires clarification to ensure the accuracy and consistency of the presented data.

    Figures 5 and 6 suffer from unclear labeling, which impedes understanding. It is essential to improve the clarity of labeling in these figures to facilitate comprehension for readers.

Addressing these issues will strengthen the clarity and comprehensibility of the manuscript, enhancing its overall quality and suitability for publication in this journal.

Reviewer 2 Report

Comments and Suggestions for Authors

line 51: delete the first name, Giovanni

line 57: put a space between the sentences. The need to insert a space occurs in several places in the manuscript

line 66: match the subject with the statement (miRNA combine)

line 75: word „unraveling” seems to be surplus, otherwise, it must be written with two l letters.

chapter 2.1 Experimental Animals: this chapter is confusing. Presents previous research with reference. There is no mention of the animals of the present investigation, it is necessary to replace them!

line 89: rephrase the first part of this sentence

line 101: I recommend using "phred score" instead of "phred"

line 102: “Excluding” with lowercase letter e

line 129: “Exact” with lowercase letter e

Figure edition: The figures were labelled with too small letters, it is necessary to increase the size.

How do the authors explain that the incidence of miRNA editing events increases at the age of 210 days?

The authors write that "Based on these findings, it is evident that the miRNA editing levels during fat development do not change smoothly but rather undergo distinct transitions." Were the above findings verified by tissue examination of the bodies of live or slaughtered animals?

How do the authors explain the age-dependent cyclicity of the intensity of fat deposition?

Accordingly, I recommend expanding the discussion.

Reviewer 3 Report

Comments and Suggestions for Authors

To the authors:

In this manuscript, the authors aim to provide a key finding related to miRNA editing during adipose tissue development in Chinese native pig, Ningxiang pigs. It effectively highlights the construction of the developmental atlas of miRNA editing and the identification of the specific editing sites at different stages depend on the growth period. Additionally, it analyzed upon the prevalence and based on the preferences of different miRNA editing, as well as significant differences in editing levels observed in the miRNA seed region. Moreover, it indicated that the potential implications of miRNA editing, particularly miR-497, in regulating fat deposition during development in pig.

However, it is important to note that while the summary captures the essence of the study, it lacks of discussion regarding the developmental insight used for miRNA editing site identification, the bioinformatics analysis performed, and the precise mechanisms through which miR-497 editing may influence fat deposition. Without these details, readers may struggle to fully grasp the significance and implications of the findings. Therefore, significant revisions and additional clarification are necessary to ensure that the study's contributions are clearly communicated to the audience.

Major points:

The authors should consider modifying the substantial revisions to the first and second major points as below:

1. The title "Construction and analysis of miRNA editing developmental atlas in adipose tissue of Ningxiang pigs" could be enhanced by incorporating terms such as "fat deposition" or "adiposity" to provide a clearer indication of the study's focus. Additionally, since Ningxiang pigs are a specific breed in China known as their ability to accumulate fat, terms like "indigenous pigs" or "native breed" will make the specific species of this pig and their relevance to the study. Incorporating these suggestions into the title would be fine that readers understand the significance of studying miRNA editing in the context of fat accumulation in this particular breed of pigs.

2. Understanding the physiological and developmental changes in pig growth from 30 to 210 days of age is crucial in swine industry. The experiments conducted the sampling at the specific ages provide scientific means with the biological processe and pathways involving miRNA at key stages of pig growth and fat development. For example, at 30 days of age, pigs are in weaning period, followed by the growth period from 90 to 150 days of age, and approaching growth maturation 180 to 210 days before slaughter. Investigating the physiological changes and fat development activity at each stage enables the swine industry to obtain valuable information for optimizing breeding management, feed formulation, disease control, and other aspects of swine production. Additionally, molecular analysis of fat development is essential for understanding the characteristics of lard type pigs, Ningxiang pig.

Specific comments:

1. Lines 43 to Line 56 contain a lengthy description regarding microRNA, which might be too complex for readers. Focusing on the involvement of miRNA in adipocyte differentiation would likely make it more understandable. Here's a revised version:

2. MiRNA is capitalized in Line 62 and Line 295. Is this a typo for 'miRNA'?

3. The difference between Figure 1A and Figure 2 is unclear. It should be clarified whether Figure 1 encompasses Figure 2 or Figure 2 needs to be clearly explained to the readers.

4. The difference between the data in Figures 1B and 1C is not clear. It would be helpful if the authors overlaid individual values within the bar graph of Figure 1C to illustrate each distribution separately, making the explanation clearer.

5. Please replace the figures with higher resolution data as the current resolution is low, and the colors of the data are too faint to read: Figure2, Figure3, Figure4, Figure 5, Figure 6, Figure Sup.1

6. Line 220 to Line223: These miRNA editing sites led to significant changes in the target gene profiles of their host miRNAs. Specifically, the editing at these sites resulted in a loss of 2,310 to 3,591 target genes, and concurrently a gain of 187 to 2,078 new target genes. (Figure 5).

It should be noted that Figure 5 should include a list of target genes in the supplement.

Round 2

Reviewer 3 Report

Comments and Suggestions for Authors

The figure and Supplemental figure could not read the names. Please modify the Figure quality. I recommend to rewrite directly the name in the Figure.
